# Integration of Chitosan and Biopesticides to Suppress Pre-Harvest Diseases of Apple

Liza DeGenring [1], Kari Peter [2] and Anissa Poleatewich [1,*]

1   Department of Agriculture, Nutrition, and Food Systems, University of New Hampshire, Durham, NH 03824, USA; liza.degenring@unh.edu
2   Department of Plant Pathology and Environmental Microbiology, Fruit Research and Extension Center, The Pennsylvania State University, Biglerville, PA 17307, USA
*   Correspondence: anissa.poleatewich@unh.edu

**Abstract:** The natural product chitosan has been shown to reduce plant disease severity and enhance the efficacy of microbial biocontrol agents in several crops. However, little is known about the potential synergisms between chitosan and biopesticides and best use practices in apple production. The objectives of this study were to evaluate the effect of pre-harvest applications of chitosan alone and in combination with a commercial biopesticide to suppress fungal diseases of apple and to investigate the potential for chitosan to reduce the quantity of overwintering *Venturia inaequalis* spores in orchard leaf litter. Chitosan products, Tidal Grow and ARMOUR-Zen 15, and a commercial biopesticide, Serenade ASO, were tested in a research orchard in Pennsylvania and commercial orchards in New Hampshire. Chitosan applications reduced apple scab incidence and severity by up to 55% on fruit compared to the water control. Chitosan also reduced sooty blotch, flyspeck, and rust incidence on fruit. Furthermore, a chitosan + biopesticide treatment overlayed onto a grower standard spray program reduced diseases more effectively than the grower standard alone. However, this efficacy was dependent on the cultivar and pathogen. Chitosan did not reduce overwintering *V. inaequalis* ascospores. This research provides evidence that pre-harvest chitosan applications have the potential for disease management in apple production.

**Keywords:** chitosan; biopesticides; natural products; apple scab; powdery mildew; *Bacillus subtilis*

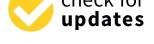



## 1. Introduction

The development and adoption of alternative products for managing plant diseases have been driven by growing concerns over the use of conventional fungicides [1,2]. Research has found that synthetic pesticides can leave chemical residues on produce, have detrimental impacts on human health, non-pest species, and the environment, and can lead to the development of fungicide resistance [3–5]. Extensive research has focused on the use of microbial biocontrol agents (BCAs) for the management of pre-harvest and post-harvest diseases over the last 30 years [6–9], resulting in the commercialization of several biopesticide products. Barriers to the widespread commercial use of BCAs include a high cost of production, limited shelf life, and variable performance under the environmental conditions found in commercial farms [10–12]. BCAs often fail to grow and maintain high enough populations in the rhizosphere and phyllosphere or do not produce antifungal compounds at levels necessary to suppress pathogen activity [13]. Finding products that enhance the efficacy of BCAs could improve disease management. Several natural compounds, such as cellulose [14], chitin [15], and chitosan [16,17], have shown a synergistic effect on biopesticide efficacy for reducing plant diseases.

Chitosan has been gaining interest as an alternative tool to promote plant growth and reduce disease. Chitosan is a natural β-(1,4)-glucosamine polymer, derived from chitin and found in insect and crustacean exoskeletons and fungal cell walls [18,19]. Chitosan

has been shown to induce a host response [20] and have fungistatic properties against several pathogenic fungi, such as *B. cinerea* (Pers.:Fr) [21,22], *Rhizopus stolonifer* (Ehrenb.:Fr.) Vuill [23], *Penicillium expansum* (Link) [22], and *Fusarium oxysporum* f. sp. *radicis-lycopersici* Jarvis and Shoemaker [24].

Much of the research evaluating chitosan as a disease management tool has focused on post-harvest applications to reduce storage rot and extend the shelf life of fruits and vegetables [25,26]. There is limited research on the pre-harvest application of chitosan to reduce plant diseases [21,27,28]. Chitosan dips for tomato seeds significantly reduced disease symptoms of *Fusarium oxysporum* f. sp. *radicis-lycopersici* and *Pythium aphanidermatum* (Edson) Fitzp. [24,29,30]. A combination of chitosan seed treatment and foliar spray on pearl millet reduced downy mildew disease caused by *Sclerospora graminicola* (Sacc.) [31]. Chitosan sprays have reduced disease caused by *B. cinerea* on petunia [21] and cucumber plants [32]. While these studies show promise for the pre-harvest application of chitosan, there are still gaps in knowledge pertaining to how efficacy is affected by molecular weight, the concentration of chitosan, solution pH, and number of applications [21,27,33].

A limited number of studies have suggested that natural compounds, such as chitin and chitosan, may have agriculturally useful synergisms when applied with a BCA [17,34]. There are several hypotheses regarding the mechanisms behind these synergisms including a heightened plant defense response [35], enhanced BCA population growth [15], and increased chitinase production by BCAs [36]. For example, *Baciullus pumilus* (PGPR strain SE 34) combined with chitosan resulted in an amplified defense response on tomatoes infected with *Fusarium oxysporum* f. sp. *radicis-lycopersici* [34].

Additionally, both chitin and chitosan have been shown to directly enhance BCAs', specifically yeasts', antagonistic activity and population growth in post-harvest applications [17,35,37]. There is some research to suggest that chitin may act as a food source for BCAs and thereby enhance biocontrol activity. Kokalis-burelle et al. reported a 60% reduction in early leafspot of peanuts when plants were treated with *Bacillus cereus* strain 304 and chitin compared to the non-treated control and a 1.3 log increase in *B. cereus* population levels compared to the non-chitin amended leaves. It was hypothesized that the chitin stimulated production of anti-fungal enzymes and helped the BCA persist long enough to compete with the pathogen by providing protection from harmful environmental variables and by providing a nutrient source [15]. These results are consistent with the effects of other food source amendments, such as cellulose [14].

Further research is needed to evaluate the efficacy of pre-harvest commercial chitosan applications and the potential synergisms of a combined application of chitosan and a BCA. Tree fruit production is an excellent model to address these knowledge gaps. Apple scab, caused by the fungus *Venturia inaequalis* (Cke.) Wint., is one of the most destructive diseases of apple (*Malus domestica* Borkh.) in the Northeast, where the moist and warm conditions during the growing season favor disease [38]. The development of fungicide resistance by *V. inaequalis* is of large concern, and several classes of fungicides, such as benzimidazole, strobilurin, and demethylation inhibitors, have already lost their effectiveness [39–41]. *Bacillus*-based biopesticides have been effective at reducing apple scab compared to a non-treated control [42,43] but typically do not achieve results comparable to conventional fungicides [44,45]. In addition to apple scab, the prevalence of powdery mildew (*Podosphaera leucotricha* (Ellis & Everh.) E.S. Salmon), cedar-apple rust (*Gymnosporangium juniperi-virginianae* Schwein.), and sooty blotch and flyspeck [46] are common diseases affecting apple production in the northeastern United States [47]. Many growers have adopted Integrated Pest Management (IPM) programs and seek alternative tools to manage diseases in their orchards. Our research aims to give growers tools to utilize in replacement of or in rotation with conventional fungicides for managing foliar diseases in apple orchards. This research is a launching point for future experiments to develop recommendations for the use of chitosan products as part of an IPM approach to manage pre-harvest diseases.



The objectives of this study were to evaluate the effects of chitosan on (1) suppressing fungal pathogens of apple when applied alone or in combination with a commercial biopesticide on a research orchard, (2) suppressing fungal pathogens of apple when applied as part of a conventional fungicide program on a commercial orchard, and (3) reducing the quantity of overwintering spores of *V. inaequalis* in orchard leaf litter.

## 2. Materials and Methods

### 2.1. Chitosan Products

The commercial chitosan product ARMOUR-Zen 15 (15% chitosan) was obtained from Botry-Zen Ltd. (Dunedin, New Zealand). The commercial chitosan product Tidal Grow (high molecular weight 2%) was obtained from Tidal Vision Inc. (Bellingham, WA, USA) (exact MWs are proprietary but are within the range of 310–375 kDa). The biopesticide, Serenade ASO, with the active ingredient *Bacillus subtilis* QST 713, was obtained from Bayer AG (Leverkusen, Germany).

### 2.2. Objective 1. Research Orchard Trials

Two experiments were conducted from 2021–2022 in a 0.8-acre research block at the Penn State University (PSU) Fruit Research and Extension Center (FREC) located in Biglerville, Pennsylvania. The orchard has an average yearly rainfall of 112 cm, average summer temperature ranging from 16 to 28 °C, and Arendtsville gravelly loam soil type [48,49]. The objective of these experiments was to evaluate the efficacy of chitosan alone or in combination with a commercial biopesticide to control pre-harvest diseases of apple. Results from 2021 were used to inform and adjust experiments in 2022. Experiments were conducted on semi-dwarf cultivar 'Rome' grafted on M.7 rootstock planted in 2015. Maintenance programs for insect pests and fire blight were applied with an airblast sprayer, delivering 100 gallon/acre to the entire orchard following commercial production practices. This research orchard had a history of apple scab and powdery mildew, and thus this research relied on natural inoculum. The Network for Environment and Weather Application's apple scab models were used to collect weather data and to predict infection periods and inoculum load using the weather station located at FREC [50].

#### 2.2.1. Experiment 1. Research Orchard Trials—2021

Five treatments were evaluated for efficacy in reducing pre-harvest diseases of apple: water control, grower standard (GS), chitosan (C), reduced risk (RR), and reduced risk + chitosan (RR + C) (Table 1). Reduced risk products are classified as having a low impact on the environment, high specificity to target organisms, and a low potential for human health risk [51,52]. The chitosan product, Tidal Grow 2%, was applied at 0.025 mg·mL$^{-1}$ (0.0025% (*v/v*) chitosan). Each treatment was applied to six replicate trees, arranged in a randomized complete block design with a buffer tree in between each treatment tree. Treatments were applied using a boom sprayer at 400 psi, delivering 100 gallon/acre. Treatment applications were made on a 7–15-day interval, starting in mid-April for primary *V. inaequalis* infection and every 10–14 days for secondary infections (Table 1). On 8 June, trees were assessed for foliar disease incidence. On 5 October, fruit were harvested and evaluated for disease incidence.

#### 2.2.2. Experiment 2. Research Orchard Trials—2022

Five treatments were evaluated in experiment 2 (Table 2). The rate of chitosan was increased from 473 mL/acre (in experiment 1) to 1893 mL/acre (0.1 mg·mL$^{-1}$ or 0.01% (*v/v*) chitosan), and the sulfur component of the reduced risk treatment in experiment 1 was removed to focus on synergisms between chitosan and the microbial biopesticide (Table 2). Each treatment was applied to five replicate trees, arranged in a randomized complete block design with a buffer tree in between each treatment tree. Treatments were applied as described for experiment 1 (Table 2). On 14 June, trees were assessed for foliar disease incidence. On 27 September, fruit were harvested and evaluated for disease incidence.

**Table 1.** Treatment list, application rate, and applicating timing for experiment 1, conducted at Penn State University Fruit Research and Extension Center in 2021. Treatments were applied to 'Rome', grafted on M.7 rootstock.

| Treatment | Trade Name (*Active Ingredient*) | Rate (per acre) | Timing [1] |
|---|---|---|---|
| Water Control | Water | -- | TC-10C |
| Grower Standard (GS) | Manzate Pro-Stick (*Mancozeb*) | 1361 g (3 lb) | TC-1C |
| | Captan Gold (*Captan*) | 1134 g (2.5 lb) | TC, 2C-10C |
| | Luna Sensation (*Fluopyram and Trifloxystrobin*) | 184 mL (5 fl oz) | P, FB |
| | Inspire Super (*Difenoconazole and Cyprodinil*) | 355 mL (12 fl oz) | PF, 1C |
| | LI 700 (*Penetrant*) | 473 mL (1 pint) | 2C-10C |
| Chitosan (C) | Tidal Grow (*2% Chitosan*) | 473 mL | TC-10C |
| Reduced Risk (RR) | Microthiol Disperss (*Sulfur 80%*) | 4536 g (10 lb) | TC-PF |
| | Serenade ASO (*Bacillus subtilis strain QST 713*) | 3785 mL (4 qt) | 1C-10C |
| Reduced Risk + Chitosan (RR + C) | Microthiol Disperss (*Sulfur 80%*) | 4536 g (10 lb) | TC-PF |
| | Serenade ASO (*Bacillus subtilis strain QST 713*) | 3785 mL (4 qt) | 1C-10C |
| | Tidal Grow (*2% Chitosan*) | 473 mL | TC-10C |

[1] Application timings: Tight Cluster (TC, 13 April); Pink (P, 27 April); Full Bloom (FB, 3 May); Petal Fall (PF, 11 May); 1st Cover (1C, 21 May); 2nd Cover (2C, 4 June); 3rd Cover (3C, 18 June); 4th Cover (4C, 30 June); 5th Cover (5C, 13 July); 6th Cover (6C, 28 July); 7th Cover (7C; 13 August); 8th Cover (8C; 26 August); 9th Cover (9C; 15 September); and 10th Cover (10C; 1 October).

**Table 2.** Treatment list, application rate, and applicating timing for experiment 2, conducted at Penn State University Fruit Research and Extension Center in 2022. Treatments were applied to 'Rome', grafted on M.7 rootstock.

| Treatment | Trade Name (*Active Ingredient*) | Rate (per acre) | Timing [1] |
|---|---|---|---|
| Water Control | Water | -- | TC-11C |
| Grower Standard (GS) | Manzate Pro-Stick (*Mancozeb*) | 1361 g (3 lb) | P-11C |
| | Captan Gold (*Captan*) | 1134 g (2.5 lb) | P |
| | Inspire Super (*Difenoconazole and Cyprodinil*) | 355 mL (12 fl oz) | P, 1C |
| | Miravis (*Pydiflumetofen*) | 101 mL (3.42 fl oz) | B. PF |
| Chitosan (C) | Tidal Grow (*2% Chitosan*) | 1893 mL | P-11C |
| Reduced Risk (RR) | Serenade ASO (*Bacillus subtilis strain QST 713*) | 3785 mL (4 qt) | P-11C |
| Reduced Risk + Chitosan (RR + C) | Serenade ASO (*Bacillus subtilis strain QST 713*) | 3785 mL (4 qt) | P-11C |
| | Tidal Grow (*2% Chitosan*) | 1893 mL | P-11C |

[1] Application timings: Tight Cluster (TC, 13 April); Pink (P, 21 April); Full Bloom (FB, 3 May); Petal Fall (PF, 11 May); 1st Cover (1C, 18 May); 2nd Cover (2C, 26 May); 3rd Cover (3C, 10 June); 4th Cover (4C, 22 June); 5th Cover (5C, 8 July); 6th Cover (6C, 20 July); 7th Cover (7C, 3 August); 8th Cover (8C, 17 August); 9th Cover (9C, 31 August); 10th Cover (10C, 12 September); and 11th Cover (11C, 23 September).

2.2.3. Objective 1. Disease Assessments

For experiments 1 and 2, trees were evaluated for incidence of five apple diseases: apple scab, powdery mildew, rust, flyspeck, and sooty blotch. In mid-June, trees were assessed for apple scab, powdery mildew, and rust incidence on leaves. For each replicate tree, foliar disease incidence was determined by randomly selecting 10 terminal shoots and evaluating all the leaves on the shoot for apple scab, powdery mildew, and rust incidence. A leaf was counted in the overall incidence of a tree if it had at least one lesion visible with the naked eye. At harvest, in early October of each year, 25 fruit per replicate tree were harvested and evaluated for incidence of apple scab, powdery mildew, rust, sooty blotch, and flyspeck. The number of apple scab lesions per fruit were counted. Additionally, 25 fruit per tree were rated for apple scab severity using a 0–6 score as described by Poleatewich et al. [42] (Figure 1A). Each of the 25 fruit were also evaluated for powdery mildew severity as a percentage of fruit showing russet symptoms using a rating scale from 0–6 (Figure 1B).

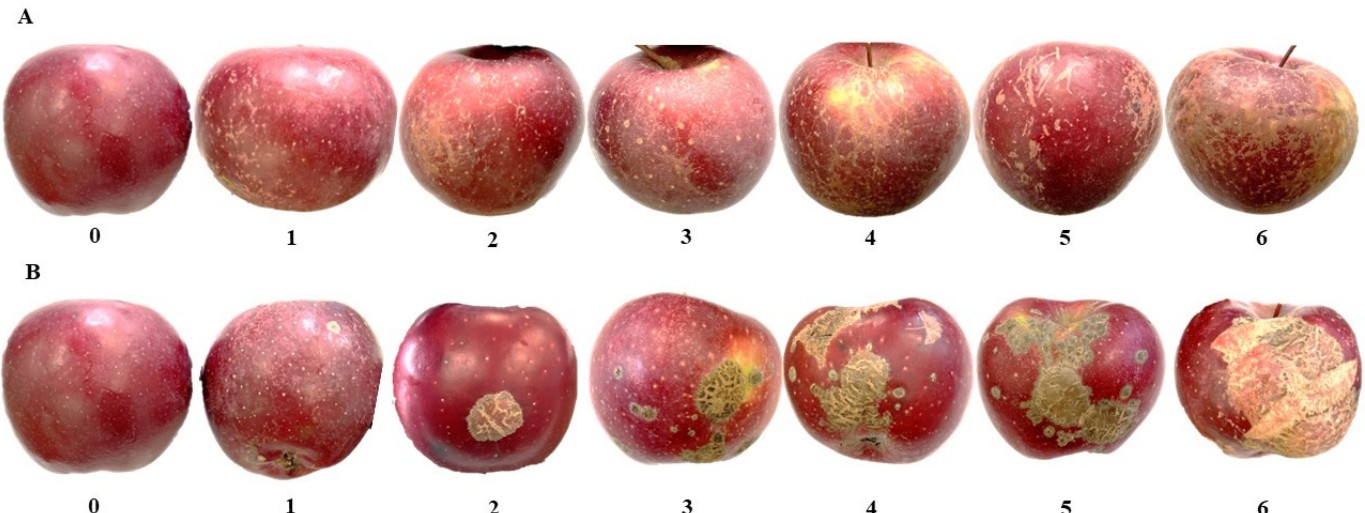

**Figure 1.** Representation for assessment of (**A**) apple russet severity and (**B**) apple scab severity on harvested fruit. Each fruit was assigned a rating (0–6) corresponding to the picture it most clearly resembled. These apples are medium sized (~8 cm in diameter).

## 2.3. Objective 2. On-Farm Trials

Two experiments were conducted in 2022 on two commercial orchard sites in New Hampshire (NH). These experiments were designed to compare disease suppression of a conventional fungicide spray program alone or in combination with the chitosan product, ARMOUR-Zen 15, applied at 1.5 mg·mL$^{-1}$ (0.15% (*v/v*) chitosan) and the biopesticide, Serenade ASO, applied at 1.5 mg·mL$^{-1}$. These farms had a history of apple scab (personal communication with farmer), and thus this research relied on natural inoculum. The NEWA's apple scab models were used to collect weather data and to predict infection periods and inoculum load using the weather station located at each site [50]).

### 2.3.1. Experiment 3. On-Farm Site #1

This trial was conducted on a commercial farm in a 3.0-acre orchard with semi-dwarf cultivars 'McIntosh' and 'Macoun', grafted on Bud.9 or M.9 rootstock, that were planted in 2017. This site has an average rainfall of 120 cm, a mean temperature of 9 °C, and a Hollis–Charlton very rocky fine sandy loam soil type [52,53]. Three treatments were evaluated: grower standard control (GS), grower standard + chitosan (GS + C), and grower standard + biopesticide + chitosan (GS + B + C) (Table 3). Each treatment was applied to seven replicate 'McIntosh' trees and eight replicate 'Macoun' trees. Treatments were applied to trees arranged in a randomized complete block design with a buffer tree in between each treatment tree. The GS treatment was applied using the grower's equipment, a Rears Pul Blast Sprayer (REARS MFG. CO., Coburg, OR, USA), delivering 38–40 gal/A. The C and B + C sprays were applied on a different day as the GS spray using two Dramm backpack BP-4Li sprayers (Model #BP-4LI and #MS40Li (MUS)) at 150 psi, delivering 23–25 gal/A. Treatment applications were made on ~10-day intervals starting on 5 May for primary *V. inaequalis* infection and ~30-day intervals for secondary infections (Table 3). The final treatment application was made on 8 September at harvest. Trees were assessed for foliar disease incidence bi-weekly starting on 7 June at symptom onset. Fruit were harvested on 8 September and put into a cold storage at 4 °C, located at the UNH Woodman Farm, until evaluations for disease incidence were initiated.

**Table 3.** Treatment list, application rates, and application timing for experiment 3, conducted in 2022. Treatments were applied to cultivars 'Macoun' and 'McIntosh'.

| Treatment | Trade Name (*Active Ingredient*) | Rate (per acre) | Timing [1] |
|---|---|---|---|
| Grower Standard (GS) | Koverall Fungicide (*Mancozeb*) | 1700 g (3.75 lbs) | GT-P |
| | Captan Gold (*Captan*) | 2612 mL (0.69 gal) | FB-1C |
| | Captan Gold (*Captan*) | 3785 mL (1 gal) | 1C-5C |
| | Agro Mos (*Copper 4%*) | 1892 mL (0.5 gal) | 3C-5C |
| Grower Standard + Chitosan (GS + C) | Koverall Fungicide (Mancozeb) | 1700 g (3.75 lbs) | GT-P |
| | Captan Gold (Captan) | 2612 mL (0.69 gal) | FB-1C |
| | Captan Gold (Captan) | 3785 mL (1 gal) | 1C-5C |
| | Agro Mos (Copper 4%) | 1892 mL (0.5 gal) | 3C-5C |
| | **ARMOUR Zen (15% Chitosan)** | **3785 mL (4 qts)** | **P-5C** |
| Grower Standard + Biopesticide + Chitosan (GS + B + C) | Koverall Fungicide (Mancozeb) | 1700 g (3.75 lbs) | GT-P |
| | Captan Gold (Captan) | 2612 mL (0.69 gal) | FB-1C |
| | Captan Gold (Captan) | 3785 mL (1 gal) | 1C-5C |
| | Agro Mos (Copper 4%) | 1892 mL (0.5 gal) | 3C-5C |
| | **Serenade ASO (Bacillus subtilis strain QST 713)** | **3785 mL (4 qts)** | **P-5C** |
| | **ARMOUR Zen (15% Chitosan)** | **3785 mL (4 qts)** | **P-5C** |

[1] Application timings based on 'McIntosh': Green Tip (GT, 22 April); Pink (P, 5 May); Full Bloom (FB, 16 May); Petal Fall (PF, 26 May); 1st Cover (1C, 7 June); 2nd Cover (2C, 20 June); 3rd Cover (3C, 22 July); 4th Cover (4C, 30 August); and 5th Cover (5C, 8 September).

2.3.2. Experiment 4. On-Farm Site #2

The second commercial orchard trial took place in a 0.9-acre orchard with semi-dwarf cider cultivars, 'Kingston Black', 'Dabinett', and 'Wickson', grafted on M.26 rootstock that were planted in 2018. This site has an average rainfall of 114 cm, an average temperature of 6 °C, and a Gilmanton fine sandy loam soil type [52,54]. Three treatments were evaluated: grower standard control (GS), grower standard + chitosan (GS + C), and grower standard + biopesticide + chitosan (GS + B + C) (Table 4). Each treatment was applied to four replicate trees, arranged in a randomized complete block design. The GS treatment was applied using the grower's equipment, a Rears powerblast sprayer (REARS MFG. CO., Coburg, OR), delivering 38–40 gal/A. The C and B + C sprays were applied on a different day as the GS spray using two Dramm backpack BP-4Li sprayers (Model #BP-4LI and #MS40Li (MUS)) at 150 psi, delivering 23–25 gal/A. Treatment applications were made on ~10-day intervals starting on 6 May for primary *V. inaequalis* infection and ~30-day intervals for secondary infections (Table 4). The final treatment application was made on 4 October at harvest. Maintenance programs for insect pests and fire blight were applied by the grower to the entire orchard following commercial production practices. Trees were assessed for foliar disease incidence monthly starting on 10 June at symptom onset. Fruit were harvested on 4 October and put into a cold storage at 4 °C until evaluations for disease incidence were initiated.

2.3.3. Objective 2. Disease Assessments

For experiments 3 and 4, trees were evaluated for incidence and severity of five apple diseases: apple scab, powdery mildew, rust, flyspeck, and sooty blotch. Apple scab severity was evaluated by using a rating scale from 0–6, as described by Poleatewich et al. [42]. Starting in mid-June, treatments were assessed for apple scab, powdery mildew, and rust incidence on leaves as described in Objective 1. Disease assessments were conducted bi-weekly for experiment 3 and monthly for experiment 4 and continued until harvest (Tables 3 and 4). For each replicate tree, foliar disease incidence was determined by randomly selecting five branches and evaluating five leaves on the branch for disease incidence.

**Table 4.** Treatment list, application rates, and application timing for experiment 4 conducted in 2022. Treatments were applied to cultivars 'Dabinett', 'Wickson', and 'Kingston Black'.

| Treatment | Trade Name (*Active Ingredient*) | Rate (per acre) | Timing [1] |
|---|---|---|---|
| Grower Standard (GS) | Kocide 3000 (Copper Hydroxide 46.1%) | 1814 g (4 lbs) | GT |
| | Koverall Fungicide (Mancozeb) | 1361 g (3 lbs) | P-PF |
| | Captan Gold (*Captan*) | 3785 mL (4 qts) | FB-3C |
| | Pristine Fungicide (*Pyraclostrobin and Boscalid*) | 454 g (16 oz) | 4C |
| Grower Standard + Chitosan (GS + C) | Kocide 3000 (Copper Hydroxide 46.1%) | 1814 g (4 lbs) | GT |
| | Koverall Fungicide (Mancozeb) | 1361 g (3 lbs) | P-PF |
| | Captan Gold (Captan) | 3785 mL (4 qts) | FB-3C |
| | Pristine Fungicide (Pyraclostrobin and Boscalid) | 454 g (16 oz) | 4C |
| | **ARMOUR Zen (15% Chitosan)** | **3785 mL (4 qts)** | **P-5C** |
| Grower Standard + Biopesticide + Chitosan (GS + B + C) | Kocide 3000 (Copper Hydroxide 46.1%) | 1814 g (4 lbs) | GT |
| | Koverall Fungicide (Mancozeb) | 1361 g (3 lbs) | P-PF |
| | Captan Gold (Captan) | 3785 mL (4 qts) | FB-3C |
| | Pristine Fungicide (Pyraclostrobin and Boscalid) | 454 g (16 oz) | 4C |
| | **Serenade ASO (Bacillus subtilis strain QST 713)** | **3785 mL (4 qts)** | **P-5C** |
| | **ARMOUR Zen (15% Chitosan)** | **3785 mL (4 qts)** | **P-5C** |

[1] Application timings based on 'Dabinett': Green Tip (GT, 18 April); Pink (P, 6 May); Full Bloom (FB, 17 May); Petal Fall (PF, 26 May); 1st Cover (1C, 10 June); 2nd Cover (2C, 21 June); 3rd Cover (3C, 19 July); 4th Cover (4C, 18 August); and 5th Cover (5C, 15 September).

At harvest, 25 fruit per replicate tree were evaluated for incidence of apple scab, powdery mildew, rust, sooty blotch, and flyspeck. A total of 25 fruit per tree were rated for apple scab severity using a 0–6 score, as described by Poleatewich et al. (Figure 1A) [43]. Each of the 25 fruit was also evaluated for powdery mildew severity as a percentage of fruit showing russet symptoms using a rating scale from 0–6 (Figure 1B).

### 2.4. Objective 3. Evaluation of Chitosan to Reduce Overwintering of V. inaequalis in Orchard Leaf Litter

The objective of this experiment was to evaluate the efficacy of chitosan to reduce overwintering ascospores of *V. inaequalis* in the orchard. Reducing *V. inaequalis* overwintering and ascospore production has been identified as an important strategy to reduce primary infections of apple scab [38]. Chitosan was compared to urea, a strategy known to reduce the overwintering inoculum of *V. inaequalis* [40,55]. Three treatments were evaluated: 5% urea solution (50 g of agriculture grade urea/L), Tidal Grow 2%, and a water control. In 2020, the application rate of the Tidal Grow 2% was 0.79 mL·L$^{-1}$ and in 2021, the application rate was 7.9 mL·L$^{-1}$. In November 2020 and 2021, apple leaves from different cultivars (i.e., 'Rome Beauty', 'Cameo', and 'Gala') infected with *V. inaequalis* were collected from a FREC orchard. The leaves were packed into sachets, which comprised two 5-inch round window screens with approximately 8 leaves placed between the screens and then stapled shut [55]. In 2020, treatments were sprayed until glistened (~0.7 mL per leaf) using a hand-held spray bottle onto 9 replicate sachets for a total of 27 sachets [55]. In 2021, treatments were sprayed onto 12 replicate sachets for a total of 36 sachets. The sachets were overwintered fixed to the orchard floor of the 5-Cultivar apple research block at FREC.

Ascospore production was evaluated weekly during the ascospore ejection period, from late April to the end of June (or until no more ascospores were observed) [38]. Three sachets per treatment were removed from the orchard each week for about twelve weeks. In the laboratory, the sachets were completely submerged in water and soaked for 1 min to initiate spore release [38]. The sachets were then placed on top of a vacuum with a slide cover at the bottom to catch spores that were released [56]. The sachets remained on the vacuum for 30 min. Once complete, the slide cover was placed on a slide bottom that contained a drop of Lactophenol blue dye. Ascospores were enumerated under the compound microscope, and the number of ascospores per sachet was determined.

*2.5. Data Analysis*

Data analysis was performed using RStudio version 2022.09.6 "Spotted Wakerobin" [57]. Each experiment was analyzed separately. Additionally, for the NH farm sites, each cultivar was analyzed separately. Disease incidence and severity ratings on fruit were transformed using an arcsine transformation to achieve homogeneity of variance. For measurements taken over time, the area under the disease progress curve (AUDPC) was calculated for each experimental unit [21,58]. Data were analyzed with an analysis of variance (ANOVA) using partial (type II) sums of squares ('car' package). Post-hoc Tukey means separation tests were conducted using least squared adjusted treatment means obtained via the 'emmeans' package in RStudio. Contrasts were also conducted but resulted in the same results as the Tukey tests. Graphs were created in RStudio using the package 'ggplot2'.

## 3. Results

*3.1. Objective 1. Experiment 1–2*

At FREC, apple scab (*V. inaequalis*), powdery mildew (*P. leucotricha*), cedar-apple rust (*G. juniperi-virginianae*), sooty blotch, and flyspeck were observed on leaves and apples. Apple scab was identified by the characteristic olive-green to brown lesions on the leaves and scabby lesions on the fruit [59]. Powdery mildew was identified by the characteristic white powdery growth on the leaves and potential leaf curling [60]. Additionally, flag shoots were observed in which an entire shoot had a silver-gray appearance because of the infection. On fruit, powdery mildew was identified by russeting and discoloration on the fruit [60]. Rust was identified by observing yellow to orange lesions on the leaves and fruit and rust projections on the underside of the leaf [61]. Sooty blotch and flyspeck are commonly observed together on harvested fruit and are characterized by sooty or cloudy blotches with indefinite borders and defined, black, shiny dots [62].

*3.2. Objective 1. Experiment 1. FREC Research Trials—2021*

During the 2021 growing season, five primary scab infection events were predicted using NEWA modeling (Supplemental Table S1). Ascospore discharge was low until 29 April when the ascospore discharge accumulation jumped from 20% to 53%. Ascospore discharge was complete by 10 May. Three infection events occurred during this two-week period, two of which lasted consecutive days (Supplemental Figure S1A).

In June, trees treated with the GS, RR, and RR + C treatments had a significantly lower incidence of apple scab on the leaves compared to the water control ($p \leq 0.011$) (Figure 2A). Trees treated with C did not have a significant reduction of foliar apple scab incidence compared to the water control ($p = 0.101$). There was no effect of treatment on powdery mildew incidence ($p = 0.206$) or rust incidence ($p = 0.103$) on the leaves (Supplemental Table S2).

In October, there was significantly less incidence and severity of apple scab on fruit harvested from trees treated with the GS, RR, and RR + C treatments compared to the water control fruit ($p \leq 0.001$) (Table 5). Fruit from these treatments had an 80% to 96% less apple scab incidence and an 82% to 97% less apple scab severity compared to the water control fruit. Additionally, all fruit from the treated trees had 61% to 99% less apple scab lesions compared to fruit from the water control trees ($p \leq 0.037$) (Table 5). Fruit harvested from the trees treated with the GS, RR, and RR + C treatments had significantly less powdery mildew incidence and russet severity compared to the water control fruit ($p \leq 0.016$) (Table 5). For flyspeck incidence, only the trees treated with GS had less disease compared to the water control ($p = 0.017$) (Table 5). There was a significantly less incidence of sooty blotch on the harvested fruit from all treatments compared to the water control ($p \leq 0.033$). Fruit from the trees treated with the GS, RR + C, and C treatments had the least amount of sooty blotch (Table 5). No fruit evaluated under any treatment showed symptoms of rust.

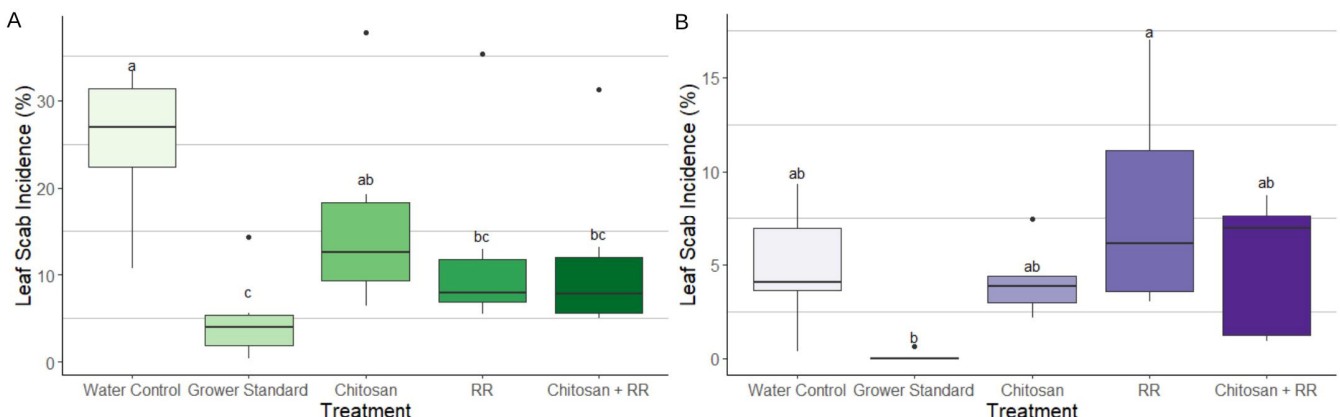

**Figure 2.** Effect of treatments on leaf scab incidence (%) on 'Rome' leaves from the (**A**) 2021 (experiment 1) and (**B**) 2022 (experiment 2) Penn State University Fruit Research and Extension Center research trials. Treatments included a water control, grower standard, chitosan, reduced risk (RR), and a chitosan + reduced risk mixture. Exact chemical treatments for each experiment are listed in Tables 1 and 2. Treatment means followed by the same letter are not significantly different ($\alpha$ = 0.05) as determined by the Tukey HSD Post-hoc test. The lines in the boxplot represent median, while the box represents the Interquartile Range (IQR). The start and end of the lines represent the minimum value in the data ($Q_3 - 1.5 \times IQR$), while any outliers are represented as dots.

**Table 5.** Mean disease incidence, mean number of scab lesions, scab severity (score 0–6), or russet severity (score 0–6) $\pm$ standard error on harvested 'Rome' fruit from the 2021 research orchard trials (experiment 1) [1,2].

| | Water Control | GS | C | RR | RR + C |
|---|---|---|---|---|---|
| Scab Incidence (%) | 56.2 ± 6.1 (a) | 2.0 ± 1.4 (b) | 39.3 ± 6.3 (a) | 6.7 ± 1.7 (b) | 11.3 ± 3.3 (b) |
| Number of Scab Lesions | 4.6 ± 0.7 (a) | 0.0 ± 0.0 (c) | 1.8 ± 0.3 (b) | 0.2 ± 0.1 (c) | 0.4 ± 0.2 (bc) |
| Scab Severity (0–6) | 0.9 ± 0.1 (a) | 0.02 ± 0.01 (b) | 0.5 ± 0.1 (a) | 0.07 ± 0.02 (b) | 0.2 ± 0.04 (b) |
| Powdery Mildew Incidence (%) | 54.6 ± 5.6 (a) | 26.0 ± 4.9 (b) | 40.0 ± 4.0 (ab) | 28.1 ± 6.1 (b) | 28.0 ± 4.0 (b) |
| Russet Severity (0–6) | 1.8 ± 0.2 (a) | 1.1 ± 0.1 (b) | 1.3 ± 0.1 (ab) | 0.8 ± 0.1 (b) | 0.9 ± 0.1 (b) |
| Sooty Blotch Incidence (%) | 57.6 ± 3.8 (a) | 14.7 ± 2.5 (c) | 32.0 ± 6.3 (bc) | 35.4 ± 5.1 (b) | 31.3 ± 6.8 (bc) |
| Flyspeck Incidence (%) | 24.5 ± 9.0 (a) | 1.3 ± 1.3 (b) | 33.3 ± 9.6 (a) | 19.2 ± 4.2 (ab) | 18.3 ± 5.2 (a) |

[1] Treatments evaluated were a water control, grower standard (GS), chitosan C), reduced risk (RR), and a reduced risk and chitosan mixture (RR + C). [2] Within a disease measurement, treatment means followed by different letters are significantly different ($\alpha$ = 0.05) as determined by the Tukey HSD Post-hoc test.

### 3.3. Objective. Experiment 2. FREC Research Trials—2022

For the 2022 season, eight primary scab infection events were predicted using NEWA modeling (Supplemental Table S1). Similar to 2021, cumulative ascospore discharge jumped from 24% to 61% on 26 April. Ascospore discharge was complete by 20 May. During these 25 days, there were six infection events, four of which lasted multiple days (Supplemental Figure S1B).

In June, only trees treated with the GS treatment had a significantly lower incidence and severity of foliar apple scab ($p \le 0.007$) (Figure 2B). Additionally, only trees treated with the GS treatment had less powdery mildew incidence compared to the water control ($p = 0.030$) (Table 6). Trees treated with GS and C had 100% and 70% less rust incidence, respectively, compared to the water control ($p \le 0.017$) (Table 6).

In October, there was a significantly less incidence and severity of apple scab on fruit harvested from trees treated with GS compared to the control ($p = 0.001$) (Table 6). Apple scab severity on the fruit from trees treated with C was 54% less compared to the water control and was as effective as the GS treatment ($p = 0.071$) (Table 6). There was no effect of treatment on powdery mildew incidence ($p = 0.873$) or severity ($p = 0.300$) on the harvested apples (Table 6). The fruit harvested from trees treated with GS and C had 100% and

92%, respectively, less rust incidence compared to the control fruit ($p \leq 0.050$) (Table 6). Similarly, fruit from the GS- and C-treated trees had 92% and 47%, respectively, less sooty blotch incidence and 100% and 68%, respectively, less flyspeck compared to the control fruit ($p \leq 0.034$) (Table 6).

**Table 6.** Mean disease incidence, mean number of scab lesions, scab severity (score 0–6), or russet severity (0–6) ± standard error on 'Rome' leaves and harvested fruit from the 2022 research orchard trial (experiment 2) [1,2].

|  |  | Water Control | GS | C | RR | RR + C |
|---|---|---|---|---|---|---|
| **Leaves** | Scab Severity (0–6) | 0.41 ± 0.07 (ab) | 0.01 ± 0.01 (b) | 0.42 ± 0.09 (ab) | 0.66 ± 0.10 (a) | 0.45 ± 0.09 (ab) |
|  | Powdery Mildew Incidence (%) | 39.0 ± 2.75 (a) | 21.1 ± 2.9 (b) | 48.9 ± 2.0 (a) | 52.3 ± 2.1 (a) | 51.2 ± 2.1 (a) |
|  | Powdery Mildew Shoot Incidence | 12.4 ± 3.9 | 7.2 ± 2.4 | 10.0 ± 1.6 | 9.8 ± 1.7 | 15.0 ± 1.2 |
|  | Rust Incidence (%) | 8.2 ± 1.3 (a) | 0.0 ± 0.0 (c) | 2.4 ± 1.8 (bc) | 5.4 ± 1.0 (ab) | 6.7 ± 1.5 (ab) |
| **Harvested Fruit** | Scab Incidence (%) | 29.6 ± 8.8 (a) | 0.8 ± 0.8 (b) | 16 ± 5.2 (a) | 24.8 ± 4.1 (a) | 39.2 ± 6.0 (a) |
|  | Number of Scab Lesions | 2.3 ± 0.7 (ab) | 0.01 ± 0.01 (b) | 2.1 ± 1.2 (ab) | 1.3 ± 0.4 (ab) | 3.6 ± 0.7 (a) |
|  | Scab Severity (0–6) | 0.62 ± 0.2 (a) | 0.01 ± 0.01 (b) | 0.28 ± 0.1 (ab) | 0.36 ± 0.1 (a) | 0.76 ± 0.2 (a) |
|  | Powdery Mildew Incidence (%) | 68.8 ± 6.5 | 72.0 ± 10.0 | 71.2 ± 7.1 | 67.2 ± 4.6 | 68.8 ± 8.2 |
|  | Russet Severity (0–6) | 1.9 ± 0.8 | 1.3 ± 0.5 | 0.9 ± 0.2 | 1.2 ± 0.08 | 1.0 ± 0.2 |
|  | Rust Incidence (%) | 9.6 ± 4.3 (a) | 0.0 ± 0.0 (b) | 0.8 ± 0.8 (b) | 11.2 ± 5.0 (a) | 4.8 ± 3.2 (ab) |
|  | Sooty Blotch Incidence (%) | 40.8 ± 3.9 (a) | 3.2 ± 0.8 (b) | 21.6 ± 6.4 (ab) | 33.6 ± 5.9 (a) | 41.6 ± 4.1 (a) |
|  | Flyspeck Incidence (%) | 20.0 ± 7.2 (a) | 0.0 ± 0.0 (b) | 6.4 ± 2.0 (ab) | 13.6 ± 6.5 (a) | 21.6 ± 4.7 (a) |

[1] Treatments evaluated were a water control, grower standard (GS), chitosan (C), reduced risk (RR), and a reduced risk and chitosan mixture (RR + C). [2] Within a disease measurement, treatment means followed by different letters are significantly different ($\alpha = 0.05$) as determined by the Tukey HSD Post-hoc test. Measurements with significant differences are visually represented by the shaded cells.

### 3.4. Objective 2. Experiment 3. NH On-Farm Site #1

For experiment 3 (on-farm site #1), conducted in 2022, six primary scab infection events were predicted using NEWA modeling (Supplemental Table S1); however, five of these events were multi-day infection periods. Ascospore discharge was steady throughout the spring, although spikes occurred around bloom and petal fall (Supplemental Figure S2A). Ascospore discharge was complete by 2 June. At this site, apple scab, powdery mildew, cedar-apple rust, sooty blotch, and flyspeck were observed on leaves and apples and were identified as discussed in Section 3.1. Additionally, frog eye leaf spot, caused by *Botryosphaeria obtuse* (Schwein.) Shoemaker, was observed and identified by the characteristic small purple flecks on the leaves that enlarge over time, resulting in a purple ring around a brown center [63].

Over the season, there was no effect of treatment on the AUDPC of scab severity on the leaves from cultivars 'Macoun' ($p = 0.789$) or 'McIntosh' ($p = 0.315$) (Table 7). However, the 'Macoun' trees treated with GS + C had a 19–25% lower AUDPC of scab incidence on their leaves compared to the GS- and GS + B + C-treated 'Macoun' trees ($p = 0.038$) (Table 7). Additionally, the 'Macoun' trees treated with GS + B + C had a 50% lower AUDPC of powdery mildew incidence compared to the GS-treated 'Macoun' trees ($p = 0.045$) (Table 7). There was no effect of treatment on the AUDPC of frog eye severity on the leaves from cultivars 'Macoun' ($p = 0.400$) or 'McIntosh' ($p = 0.181$) (Table 7). However, the 'McIntosh' trees treated with GS and GS + B + C had a significantly lower AUDPC of frog eye incidence on their leaves compared to the GS + C-treated trees ($p = 0.050$) (Table 7). Rust severity and incidence were not affected by treatment on the 'Macoun' trees ($p \geq 0.503$), and the 'McIntosh' trees had only a few leaves with rust lesions (Table 7).

In September, the 'McIntosh' trees treated with GS + B + C had a 33–35% lower scab incidence and 64–79% less scab lesions on the harvested fruit compared to the 'McIntosh' trees treated with GS or GS + C ($p \leq 0.048$) (Table 7). However, there was no effect of treatment on apple scab severity for the fruit harvested from cultivars 'Macoun' ($p = 0.634$) or 'McIntosh' ($p = 0.176$) (Table 7). While there was no effect of treatment on powdery mildew incidence or severity on the 'Macoun' fruit ($p \geq 0.605$), the 'McIntosh' fruit from trees treated with GS + C had a 34% less powdery mildew incidence and 65% less powdery mildew severity compared to fruit treated with GS ($p \leq 0.015$) (Table 7). There was no effect of treatment on sooty blotch or flyspeck incidence for the fruit harvested from cultivars

'Macoun' ($p \geq 0.320$) or ''McIntosh ($p \geq 0.142$) (Table 7). No fruit evaluated under any treatment or cultivar showed symptoms of rust.

**Table 7.** Effects of grower standard control (GS), grower standard + chitosan (GS + C), and grower standard + biopesticide + chitosan (GS + B + C) on the area under the disease progress curve (AUDPC) [1] for foliar disease incidence or severity ± standard error on cultivars 'Macoun' and 'McIntosh' from the 2022 New Hampshire on-farm trial #1. Mean disease incidence, mean number of scab lesions, scab severity (score 0–6), or russet severity (score 0–6) ± standard error on harvested apples from experiment 3 conducted in 2022 [2].

|  |  | Macoun | | | McIntosh | | |
|---|---|---|---|---|---|---|---|
|  |  | **GS** | **GS + C** | **GS + B + C** | **GS** | **GS + C** | **GS + B + C** |
| **Leaves** | Scab Incidence | 45.5 ± 8.7 (ab) | 36.5 ± 9.2 (a) | 49.0 ± 7.4 (b) | 226.0 ± 26.0 | 213.7 ± 43.7 | 236.3 ± 22.4 |
|  | Scab Severity | 1.42 ± 0.7 | 1.1 ± 0.4 | 3.9 ± 0.6 | 2.8 ± 0.7 | 3.8 ± 1.0 | 1.2 ± 0.4 |
|  | Powdery Mildew Incidence | 54.0 ± 18.5 (a) | 37.5 ± 7.9 (ab) | 27.5 ± 5.9 (b) | 7.1 ± 1.8 | 6.3 ± 1.5 | 9.4 ± 5.5 |
|  | Rust Incidence | 67.0 ± 23.2 | 74.5 ± 26.3 | 73.25 ± 27.0 | 0.0 ± 0.0 | 0.0 ± 0.0 | 1.7 ± 0.8 |
|  | Rust Severity | 1.4 ± 0.5 | 2.0 ± 0.6 | 1.6 ± 0.6 | 0.01 ± 0.01 | 0.01 ± 0.01 | 0.05 ± 0.05 |
|  | Frog Eye Incidence | 148.5 ± 41.1 | 129.5 ± 32.1 | 132.0 ± 30.0 | 42.3 ± 10.4 (b) | 104.6 ± 35.2 (a) | 48.0 ± 9.9 (b) |
|  | Frog Eye Severity | 5.6 ± 2.0 | 3.8 ± 1.4 | 3.9 ± 1.3 | 1.0 ± 0.3 | 0.8 ± 0.1 | 3.1 ± 1.5 |
| **Harvested Fruit** | Scab Incidence (%) | 50.4 ± 14.1 | 46.3 ± 10.1 | 46.3 ± 15.8 | 57.5 ± 15.8 (a) | 56.0 ± 16.3 (a) | 37.6 ± 5.2 (b) |
|  | Number of Scab Lesions | 1.4 ± 0.2 | 1.2 ± 0.2 | 0.9 ± 0.1 | 3.9 ± 0.7 (ab) | 6.6 ± 1.2 (a) | 1.4 ± 0.4 (b) |
|  | Scab Severity (0–6) | 0.9 ± 0.1 | 0.7 ± 0.1 | 0.6 ± 0.1 | 1.1 ± 0.1 | 1.1 ± 0.1 | 0.5 ± 0.1 |
|  | Powdery Mildew Incidence (%) | 68 ± 9.2 | 68.6 ± 4.6 | 75.5 ± 8.7 | 90.2 ± 2.4 (a) | 59.6 ± 7.5 (b) | 78.9 ± 4.1 (ab) |
|  | Russet Score (0–6) | 1.1 ± 0.1 | 1.0 ± 0.1 | 1.6 ± 0.2 | 2.0 ± 0.2 (a) | 0.7 ± 0.1 (b) | 2.0 ± 0.2 (a) |
|  | Sooty Blotch Incidence (%) | 1.6 ± 1.6 | 4.0 ± 1.7 | 1.0 ± 1.0 | 1.3 ± 1.3 | 0.8 ± 0.8 | 0.8 ± 0.8 |
|  | Flyspeck Incidence (%) | 19.2 ± 5.4 | 18.3 ± 4.4 | 15.0 ± 6.0 | 8.0 ± 5.4 | 8.8 ± 4.0 | 23.9 ± 9.7 |

[1] The AUDPC was calculated from disease assessments taken bi-weekly starting on 7 June until 15 September.
[2] Within a disease measurement and within a cultivar ('Macoun' and 'McIntosh'), treatment means followed by different letters are significantly different ($\alpha$ = 0.05) as determined by the Tukey HSD Post-hoc test. Measurements with significant differences are visually represented by the shaded cells.

### 3.5. Objective 2. Experiment 4. NH On-Farm Trial #2

For experiment 4 (on-farm site #2), seven primary scab infection events were predicted using NEWA modeling (Supplemental Table S1), all of which were multi-day infection periods. Cumulative ascospore discharge jumped from 28% to 50% around bloom. Ascospore discharge was complete by 6 June. During these 22 days, there were four infection events (Supplemental Figure S2B).

Apple scab was not observed on any trees at NH on-farm site #2. Overall, disease incidence was low at this farm, so incidence and severity data for all foliar diseases was combined to evaluate the effects of treatments on the suppression of all leaf and fruit spots. Over the season, there was no effect of treatment on the AUDPC of disease severity on leaves from cultivars 'Dabinett' ($p$ = 0.355), 'Kingston Black' ($p$ = 0.202), or 'Wickson' ($p$ = 0.912) (Table 8). However, 'Kingston Black' trees treated with GS + C had a 55% lower AUDPC of disease incidence compared to 'Kingston Black' trees treated with GS ($p$ = 0.049) (Table 8). Additionally, 'Kingston Black' trees treated with GS + B + C had a significantly lower AUDPC of powdery mildew incidence compared to 'Kingston Black' trees treated with GS ($p$ = 0.038) (Table 8). There was no effect of treatment on AUDPC of frog eye lesion severity on leaves from cultivars 'Dabinett' ($p$ = 0.235) or 'Wickson' ($p$ = 0.810) (Table 8). However, 'Kingston Black' trees treated with GS or GS + C had a 68% and 88%, respectively, lower AUDPC of frog eye severity compared to 'Kingston Black' trees treated with GS + B + C ($p \leq 0.022$) (Table 8).

In October, there was no effect of treatment on incidence or severity of powdery mildew or flyspeck on the fruit harvested from cultivars 'Dabinett' ($p \geq 0.183$) or 'Wickson' ($p \geq 0.133$) (Supplemental Table S3). Cultivar 'Kingston Black' did not produce enough fruit for post-harvest evaluations. No fruit evaluated under any treatment or cultivar showed symptoms of rust or sooty blotch.

**Table 8.** Effects of grower standard control (GS), grower standard + chitosan (GS + C), and grower standard + biopesticide + chitosan combination (GS + B + C) on area under the disease progress curve (AUDPC) [1] for disease incidence or severity ± standard error on cultivars 'Kingston Black', 'Dabinett', and 'Wickson' leaves from experiment 4 in 2022 [2].

| | Kingston Black | | | Dabinett | | | Wickson | | |
| --- | --- | --- | --- | --- | --- | --- | --- | --- | --- |
| | GS | GS + C | GS + B + C | GS | GS + C | GS + B + C | GS | GS + C | GS + B + C |
| Powdery Mildew Incidence | 45.5 ± 11 (ab) | 61.5 ± 20 (a) | 24.0 ± 11 (b) | 10.0 ± 2.7 | 17.5 ± 6.2 | 27.0 ± 10 | 13.0 ± 6.0 | 10.0 ± 5.0 | 13.0 ± 4.7 |
| Lesion Incidence [3] | 53.0 ± 12 (a) | 24.0 ± 1.6 (b) | 31.0 ± 4.4 (ab) | 6.0 ± 2.6 | 11.0 ± 6.0 | 18.0 ± 4.8 | 4.0 ± 1.6 | 9.0 ± 4.1 | 11.0 ± 3.4 |
| Lesion Severity | 4.8 ± 1.7 | 2.0 ± 0.7 | 1.5 ± 0.7 | 0.1 ± 0.1 | 0.2 ± 0.1 | 0.3 ± 0.1 | 0.1 ± 0.0 | 0.2 ± 0.1 | 0.2 ± 0.0 |
| Frog Eye Incidence | 18.0 ± 6.6 (ab) | 9.5 ± 2.4 (b) | 37.0 ± 13.3 (a) | 7.5 ± 1.9 | 13.0 ± 2.6 | 9.0 ± 1.3 | 6.5 ± 3.9 | 5.0 ± 2.1 | 8.0 ± 2.8 |
| Frog Eye Severity | 0.3 ± 0.1 (b) | 0.1 ± 0.0 (b) | 0.9 ± 0.2 (a) | 0.1 ± 0.1 | 0.4 ± 0.1 | 0.2 ± 0.0 | 0.1 ± 0.0 | 0.1 ± 0.0 | 0.1 ± 0.1 |

[1] The AUDPC was calculated from disease assessments taken monthly starting on 10 June until 4 October. [2] Within a disease measurement and within a cultivar ('Kingston Black', 'Dabinett' and 'Wickson'), treatment means followed by different letters are significantly different ($\alpha$ = 0.05) as determined by the Tukey HSD Post-hoc test. Shaded rows represent measurements with significant differences. [3] Lesion incidence represents the observation of any foliar disease on a leaf within a disease assessment.

### 3.6. Objective 3. Evaluation of Chitosan to Reduce Overwintering of V. inaequalis in Orchard Leaf Litter

In the 2020–2021 overwintering trials, the 5% urea application significantly reduced the AUDPC of ascospore production ($p < 0.001$) compared to the control, but the chitosan application did not reduce AUDPC ($p = 0.996$) (Figure 3A). In the 2021–2022 overwintering trials, even with the increase in the percent chitosan, the same results were observed: the chitosan application did not reduce the AUDPC of ascospore production ($p = 0.851$), but the 5% urea application significantly reduced the AUDPC of ascospore production compared to the control ($p < 0.001$) (Figure 3B).

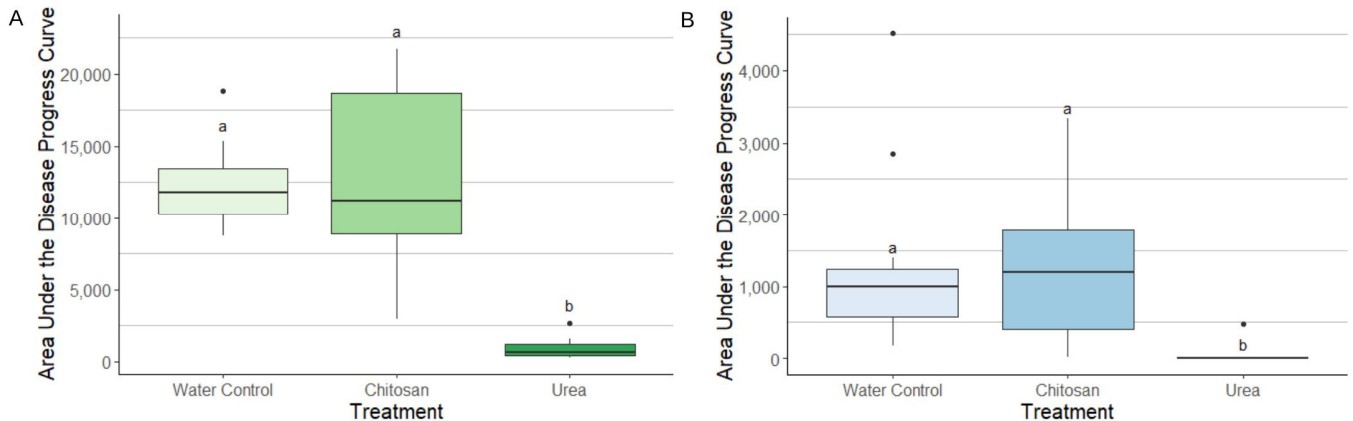

**Figure 3.** Effects of overwintering treatments on the area under the disease progress curve (AUDPC) for *Venturia inaequalis* ascospore release starting in March until mid-Summer for (**A**) 2020–2021 trial and (**B**) 2021–2022 trial at Penn State University Fruit Research and Extension Center. Treatments were a water control, chitosan (2020/2021: 0.79 mL/L chitosan; 2021/2022: 7.9 mL/L chitosan), and a 5% urea solution. Treatment means followed by the same letter are not significantly different ($\alpha$ = 0.05) as determined by the Tukey HSD Post-hoc test. The lines in the boxplot represent the median, while the box represents the Interquartile Range (IQR). The start and end of the lines represent the minimum value in the data ($Q_3 - 1.5 \times IQR$), while any outliers are represented as dots.

## 4. Discussion

The addition of BCAs and chitosan to an IPM program to manage diseases in northeast apple orchards would give growers additional disease control options. This study investigated the use of chitosan in combination with biopesticides, as part of a conventional spray program and as a tool to reduce primary apple scab infections.

### 4.1. Chitosan Can Reduce Disease When Applied as Part of a Conventional Fungicide Program

The results from this research suggest that chitosan may not be effective as a stand-alone treatment and more research is needed to determine if higher concentrations of chitosan are effective. Over the two-year study at the FREC research orchard, two chitosan concentrations were tested. The chitosan concentration was increased in year two (from $0.02$ mg·mL$^{-1}$ to $0.1$ mg·mL$^{-1}$) due to the low efficacy observed in the first year and previous reports that a higher chitosan concentration results in increased disease suppression. Specifically, chitosan concentrations between $1.0$–$2.5$ mg·mL$^{-1}$ tend to correlate with greater fungal growth inhibition compared to lower concentrations ($0.1$–$0.75$ mg·mL$^{-1}$) [17,21,23,24,64]. The low incidence of apple scab in 2022 (0–10%) made determining statistical differences between treatments difficult. Thus, it is challenging to conclude that the chitosan rate tested in 2022 was more effective than the rate tested in 2021. Felipini et al. observed a reduction in foliar apple scab with reagent grade chitosan applied at $0.5$ mg·mL$^{-1}$ compared to a control in a greenhouse with artificially inoculated leaves [65]. Future work is also needed to determine costs per acre of chitosan to identify the most effective and economically feasible chitosan concentrations and should be repeated in years when the disease pressure is higher.

At the on-farm trial sites, disease suppression by the GS + C treatment was often not significantly different from the GS treatment. However, there were instances in which the GS + C treatments had a significantly greater reduction in disease compared to the GS treatment alone. This was specifically prevalent in fruit harvested from the trees treated with GS + C. For example, the 'McIntosh' and 'Dabinett' fruit treated with GS + C had a 65% lower powdery mildew severity and a 62% less flyspeck incidence, respectively, compared to the GS-treated fruit. Data from this study suggests that chitosan can enhance the efficacy of conventional fungicide programs but may vary by cultivar. More research is needed to evaluate chitosan on other rootstock and scion cultivar combinations.

Variability in our results may be explained by the modes of action of chitosan which could be heavily influenced by the environment and plant genetics. For example, chitosan has been shown to activate plant defenses, thus indirectly reducing disease [66]. The magnitude and effectiveness of the defense response, however, is known to be dependent on plant genotype [67]. Chitosan elicits a host response characterized by increased enzymatic activity of chitinase and β-1,3-glucanases, up-regulation of the production of plant-defense related enzymes, such as polyphenoloxidase (PPO) and peroxidase (POD), and enhanced production of callose cell wall appositions in the host's epidermis and outer cortex [34,68–72]. In-vitro assays have also shown that chitosan can have a direct inhibition of fungal mycelium growth and conidial germination and elongation [21,23,73]. Additionally, research on post-harvest applications suggests that chitosan can have direct antimicrobial activity through causing cellular disorganization of the fungal pathogens [34,74,75]. For direct inhibition to occur, pathogens must come into contact with chitosan. During the season, rain events may wash chitosan away, rendering it less effective. In this study, chitosan was applied on a schedule based on predicted infection events. Applying chitosan on a calendar schedule or with a spreader sticker product may lengthen the presence of chitosan on the leaves and fruit. Additional research is needed to determine the ideal timing of chitosan application with respect to predicted infection events. Additionally, this research was limited to commercial chitosan products available in the United States, but there are other chitosan products available around the world [76] and technologies, such as chitosan nanoparticles [77,78], that can be evaluated for foliar disease management.

### 4.2. Synergisms between Chitosan and Biopesticides Varied by Site, Cultivar, and Pathogen

In this research, we tested the biopesticide Serenade ASO (*B. subtilis* QST 713). *Bacillus* BCAs have been shown to be effective at reducing apple scab compared to a non-treated control [42,43]. However, many studies have found that stand-alone applications of *Bacillus* BCAs are not comparable to a fungicide application [43–45]. Thus, combining *Bacillus* BCAs with other products or in rotation with fungicides may be a more effective strategy

for managing apple scab [42,79]. To our knowledge, no research has been conducted on the combined application of *Bacillus* BCAs and chitosan for management of pre-harvest apple diseases.

Results from the research orchard experiments suggest that the addition of chitosan to the RR treatment was not synergistic, as there were no significant differences between the RR + C and RR treatments with respect to a reduction in disease. This lack of synergism may be mainly due to the low efficacy of this chitosan product observed in our research trials. As previously discussed, the optimal chitosan concentration and application timing still need to be investigated. The chitosan product may have washed off before it was able to provide a food source for the BCA or stimulate antimicrobial enzymes. Very few studies have examined the co-application of chitosan and a BCA under field conditions [17,34–36]. Kokalis-Burelle et al. found that the co-application of reagent grade chitin and *B. cereus* along with a spreader-sticker (SoyDex) enhanced disease suppression, likely due to improved foliar colonization of the BCA [15]. Future research is needed to evaluate commercially available chitosan products applied with a surfactant as a strategy to reduce variability.

The results from the on-farm sites suggest that B + C overlayed onto a conventional fungicide program can improve disease suppression for management of apple scab and powdery mildew. The GS + B + C treatment exhibited enhanced disease suppression compared to the GS treatment. However, this efficacy was dependent on the cultivar and fungal pathogen. For example, the GS + B + C resulted in up to a 50% suppression of powdery mildew on the 'Macoun' leaves (but not fruit) and the 'McIntosh' fruit (but not leaves) compared to the GS treatment. Overall, the 'McIntosh' fruit from the GS + B + C-treated trees had a 35% and 64% lower AUDPC of scab incidence and number of scab lesions, respectively, compared to the apples from the GS-treated trees. While 35% AUDPC of apple scab is not ideal for growers, it is significantly better than the 57% AUDPC of apple scab incidence observed on the GS-treated fruit. 'Kingston Black' trees treated with GS + B + C had a 47% and 42% lower AUDPC for powdery mildew incidence and general lesion incidence. However, for many of the fungal pathogens and cultivars, there was no effect of treatment on AUDPC. Differences in efficacy of chitosan observed in the on-farm and research orchard sites may be due to the use of two different chitosan products containing different forms of chitosan (extraction method, molecular weight, and formulation). Thus, the greater efficacy of the B + C + GS treatments at the on-farms sites may be that ARMOUR-Zen was more effective at reducing disease or enhancing the BCA compared to Tidal Grow. Additional experiments directly comparing these two products are needed to make more definitive conclusions. More research is also needed to test synergisms between chitosan and other BCAs, including different strains of bacterial and fungal-based biopesticides. *Cladosporium cladosporioides* H39 [80] and *Microphaeropsis* sp. [81] have shown efficacy against apple scab and could be examined in combination with chitosan.

### 4.3. Chitosan Did Not Reduce Overwintering Spores of V. inaequalis

In the northeastern United States, ascospores produced on diseased leaves in the leaf litter constitute the primary inoculum causing apple scab [38]. Shredding leaf litter with a flail mower and/or treating the leaf litter with urea are common sanitation practices to reduce ascospore production in the orchard [40,55]. We hypothesized that chitosan would reduce overwintering ascospores through promoting microbial activity, leading to enhanced decomposition of leaves and reduction in maturation of pseudothecia [38,82,83]. Additionally, the chitosan spray on the leaves could have resulted in a direct antimicrobial activity against the *V. inaequalis* pseudothecia on the leaf. Fall chitosan applications, at the rates tested, did not reduce ascospore production in the spring. Previous research focused on optimizing urea application timing [55,84] could be applied to future trials conducted on chitosan, focusing on application rates and timing to better understand if chitosan could play a role in reducing overwintering *V. inaequalis* ascospores.

## 5. Conclusions

While not as effective compared to a conventional fungicide program, pre-harvest applications of chitosan significantly reduced apple diseases when applied alone or in combination with a biopesticide or a conventional fungicide program. The results also indicate that the application rates of the chitosan tested do not reduce overwintering ascospores. This research provides evidence that chitosan has potential as an IPM tool, but more research is needed to determine best practices for its use in an integrated management program for control of diseases in apple production in the northeast. Additional research is also needed to identify factors influencing variability and to study co-application of chitosan and other BCAs, including different strains of bacterial and fungal-based biopesticides, for managing diseases. Due to the exploratory nature of this work, sample sizes were small, and trials were conducted in a limited geographical location. Once an optimal chitosan rate and strategy for integration with other IPM products is determined, large scale trials should be conducted.

**Supplementary Materials:** The following supporting information can be downloaded at: https://www.mdpi.com/article/10.3390/horticulturae9060707/s1, Figure S1: Cumulative *Venturia inaequalis* ascospore discharge and primary scab infection events (of more than one day) data from Network for Environment and Weather Application's apple scab models, based on data collected from the weather station located at Penn State University Fruit Research and Extension Center: (A) experiment 1 and (B) experiment 2. Figure S2: Cumulative *Venturia inaequalis* ascospore discharge and primary scab infection events (of more than one day) data from Network for Environment and Weather Application's apple scab models, based on data collected from the weather station located at New Hampshire sites: (A) experiment 3 and (B) experiment 4. Table S1: Apple scab infection events and rainfall collected from Network for Environment and Weather Application's apple scab models, based on data collected from the weather station located at each site [50]. Table S2: Mean disease incidence ± standard error on 'Rome' leaves from the 2021 Penn State University Fruit Research and Extension Center research trial (experiment 1). Table S3: Mean disease incidence or russet severity (score 0–6) ± standard error on 'Dabinett' and 'Wickson' harvested apples from the 2022 New Hampshire on-farm site #2.

**Author Contributions:** Conceptualization, L.D. and A.P.; methodology, L.D., K.P. and A.P.; software, L.D.; validation, K.P. and A.P.; formal analysis, L.D.; investigation, L.D. and K.P.; resources, L.D., K.P. and A.P.; data curation, L.D.; writing—original draft preparation, L.D. and A.P.; writing—review and editing, L.D., K.P. and A.P.; visualization, L.D.; supervision, A.P.; project administration, A.P.; funding acquisition, L.D., K.P. and A.P. All authors have read and agreed to the published version of the manuscript.

**Funding:** This material is based upon work supported by the National Institute of Food and Agriculture (NIFA), the U.S. Department of Agriculture (USDA), through the Northeast Sustainable Agriculture Research and Education program, under subaward number GNE19-198-33243. Additionally, funding for this project was made possible by the USDA Agricultural Marketing Service, through grant AM190100XXXXG012, as well as by the USDA NIFA Federal Appropriations, under Project PEN04694 Accession 1018736. The contents of this article are solely the responsibility of the authors and do not necessarily represent the official views of the USDA.

**Data Availability Statement:** The data that support the findings of this study are available on request from the corresponding author.

**Acknowledgments:** The authors thank the National Institute of Food and Agriculture and the U.S. Department of Agriculture for their support of this research through the Northeast Sustainable Agriculture Research and Education Production program. Additional support was provided by the University of New Hampshire College of Life Sciences and Agriculture and the New Hampshire Agricultural Experiment Station. Special thanks go to Cheryl Smith for her insight and support in this research. Thank you to the two commercial orchards and their farm managers for supporting this research and donating apples for disease assessments, post-harvest. Thank you to the technicians who worked on this research: at UNH, Allie Wilford, Cameron Mehalek, and Martina Florian; and at PSU, Brian Lehman, Teresa Krawczyk, Kate Thomas, Jordyn Hartsock, Luke May, Carl Bower,

Cody Kime, and Jared Shelly. Research was conducted by the first author in partial fulfillment of the requirements for the PhD degree of Agricultural Science, University of New Hampshire.

**Conflicts of Interest:** The authors declare no conflict of interest.

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
