# Peer review of "Integration of Chitosan and Biopesticides to Suppress Pre-Harvest Diseases of Apple"

_horticulturae, doi:10.3390/horticulturae9060707_

Round 1

Reviewer 1 Report

horticulturae-2425957: I have carefully read this manuscript, and considered it as the interesting study. However, some issues must be improved as follow:

1) Lines 28-29: What are the impact of using chemical substance to eliminate the diseases? Please provide more details and examples. You can said that “using chemical substance can harm the human health [4], increase the environmental impact [5], and reduce farmers' returns [6].” Please see these papers. {[4] Environmental, Human and Ecotoxicological Impacts of Different Rice Cultivation Systems in Northern Thailand. Int. J. Environ. Res. Public Health 2023, 20(3), 2738.} {[5] Carbon, Nitrogen and Water Footprints of Organic Rice and Conventional Rice Production over 4 Years of Cultivation: A Case Study in the Lower North of Thailand. Agronomy 2022, 12(2), 380.} {[6] Why farmers continue to use pesticides despite environmental, health and sustainability costs. Ecological Economics 2001.39, 3, 449-462.}

2) Lines 91-94: The contribution of this study should be mentioned.

3) The experiment sites must be provide the weather and soil conditions.

4) Figure 1: If possible the scale should be presented. Because it is difficult to imagine how large?

5) Figure 3: What is the unit of Y axis?

6) Line 449: Why more research is needed in Northeast? Please explain more details.

7) Conclusions should be revised by concluding based on the objectives.

Minor editing of English language required.

Author Response

1) Lines 28-29: What are the impact of using chemical substance to eliminate the diseases? Please provide more details and examples. You can said that “using chemical substance can harm the human health [4], increase the environmental impact [5], and reduce farmers' returns [6].” Please see these papers. {[4] Environmental, Human and Ecotoxicological Impacts of Different Rice Cultivation Systems in Northern Thailand. Int. J. Environ. Res. Public Health 2023, 20(3), 2738.} {[5] Carbon, Nitrogen and Water Footprints of Organic Rice and Conventional Rice Production over 4 Years of Cultivation: A Case Study in the Lower North of Thailand. Agronomy 2022, 12(2), 380.} {[6] Why farmers continue to use pesticides despite environmental, health and sustainability costs. Ecological Economics 2001.39, 3, 449-462.}

This is a good suggestion, and the text has been edited to include: Research has found that synthetic pesticides can leave chemical residues on produce, have detrimental impacts on human health, non-pest species, and the environment, and can lead to the development of fungicide resistance.

2) Lines 91-94: The contribution of this study should be mentioned.

We agree and sentences were added to highlight the contributions of this study.

3) The experiment sites must be provide the weather and soil conditions.

The weather data (rainfall and weather events conducive to disease establishment) are provided in the supplementary materials. Additionally, average weather conditions and soil types were included for all three experiment sites.  

4) Figure 1: If possible the scale should be presented. Because it is difficult to imagine how large?

These apples are medium sized apples whose diameter is around 8 cm in length. This wording was added to the Figure caption to give a better idea of the size of the apples in the image.

5) Figure 3: What is the unit of Y axis?

There is no unit for the Y axis. The area under the disease progress curve is a quantitative summary of disease intensity over time that is calculated. See this article for confirmation that the AUDPC axis do not have a unit associated with it: https://doi.org/10.1094/PHYTO-02-18-0047-R

6) Line 449: Why more research is needed in Northeast? Please explain more details.

The goal of this study was to investigate chitosan in the northeastern region of the United States. Our results suggest that chitosan may be effective, but more work is needed to identify application rates before growers in the region can use the product. While results from this study could be used throughout the world, each region has unique climatic conditions that may require different strategies. For example, the pacific northwestern U.S. apple growing regions are quite dry compared to the northeastern region.

7) Conclusions should be revised by concluding based on the objectives.

We modified the section headers in the discussion to state the main take-aways of the research more clearly as it relates to the three objectives.

Reviewer 2 Report

Overall this is a well written manuscript, as well as being interesting research but difficult to follow, it presents inputs in the combination of chemicals and membranes made with chitosan in combination with Bacillus subtilis (Not in all cases), so it is difficult to understand, and analyze. I also consider the original work, however, it requires some major changes for it to be published.

Changes to be made:

In the section:

1. Introduction, information regarding the disease caused by Venturia inaequalis in the apple crop must be attached, mentioning symptoms and product losses.

Delete the marked part in the document, we think is another interesting topic, however, outside the place of the main investigation. Better explain in more detail what apple scab is and mention the advantages of microbial biological control agents with respect to the environment and health.

Explain in detail the other diseases discussed in the results.

2. Materials and Methods

Section 2.2.

Line 116, attach reference where the identification of the pathogen or development of the disease is mentioned.

Line 147, attach reference.

Line 179, attach reference where the identification of the pathogen or development of the disease is mentioned.

Line 235, in image 1 it says 6 degrees and here it refers to 7, correct.

line 258, attach reference and explain this methodology in more detail.

Line 269, attach the number of observations and the ranges used.

The experimental design and the repetitions used in the different experiments are not the same. It is difficult to understand the methodology for the different experiments presented, together with 5 diseases, likewise the statistical test used is not the most appropriate, since it takes into account the individuality of the experiments.

Results

It is recommended to attach photos for each of the diseases registered in its evaluation scale, in addition to carrying out a statistical analysis comparing the different varieties of apple trees and delimiting which is the most susceptible.

The most vulnerable aspect of the study is that there is no precise identification of the 5 casal agents of the disease, as well as if virulent strains are handled. It is recommended to carry out the molecular identification process to corroborate the taxonomy of the causal agents.

To this is added, that the best treatment presented was chitosan plus a biopesticide or program with a chemical agent. This is counterproductive since the chemical agent penetrates the chitosan membrane and is a potential risk to the health of consumers, in addition to the fact that it can increase the production costs of the apple tree by removing this membrane. However, more studies would be necessary to know the percentage of the product that is present in the apple tree.

Chemicals like Mancozeb have been shown to cause health problems and are linked to cancer problems in farmers.

In the methodology the microscopic observation of ascospores is mentioned and in the results no image is presented, please attach photographic information.

Discussion

The blue color, go to conclusion.

Adequate, however, it is recommended to specify the information more, since the authors refer to the disease, but since there are 5 different diseases, in some paragraphs they do differ from the disease, but in most of the discussion they do not, please restructure the justification so that it can be more understandable.

Conclusion

It is recommended to substantially modify the conclusion according to the results presented.

Bibliography

Adequate

However modify:

Color yellow, italicize

Red color, lowercase

Author Response

1. Introduction, information regarding the disease caused by Venturia inaequalis in the apple crop must be attached, mentioning symptoms and product losses.

We agree that this information is important, however the focus of this research was to determine if chitosan can be used to manage foliar and fruit diseases in apple orchards and not on any one pathogen. Thus, our introduction focuses on research and gaps in knowledge about chitosan and the combination of chitosan and biopesticides for disease management. We intentionally did not focus on the biology of foliar diseases or their epidemiology as it would have taken away from the main objective of our paper. We cite papers that provide supporting evidence that the five pathogens evaluated in this study are important and provide background info on the disease cycles, symptoms, and resulting product loss.

2. Delete the marked part in the document, we think is another interesting topic, however, outside the place of the main investigation. Better explain in more detail what apple scab is and mention the advantages of microbial biological control agents with respect to the environment and health.

Since this research was not exclusively focused on apple scab, we included this text to provide the reader with an overview of the many diseases affecting apple production in the northeastern US. We revised this section to focus more on the diseases of importance and avoid introducing an unrelated topic.

3. Explain in detail the other diseases discussed in the results.

See two previous comments.

Materials and Methods

Section 2.2.

4. Line 116, attach reference where the identification of the pathogen or development of the disease is mentioned.

Line 147, attach reference.

Line 179, attach reference where the identification of the pathogen or development of the disease is mentioned.

Regarding these three comments, we are unclear what is being requested as the given line numbers do not have text that refers to the ID of the pathogen. We utilized natural inoculum and thus do not have references for strains used in the research. Assessments were made based on symptoms and the causal agents were not isolated for identification. However, text was added in the results section to indicate how each disease was identified based on diagnostic symptomology.

5. Line 235, in image 1 it says 6 degrees and here it refers to 7, correct.

Changed to say 0-6.

6. line 258, attach reference and explain this methodology in more detail.

A reference was added to this methodology. Additionally, details were added to the sentence to give more details.

7. Line 269, attach the number of observations and the ranges used.

Additional details were added to this section to make the methodology clearer. Sachets were collected weekly for a total of 12 observations. At each collection time, 3 replicate sachets were collected and the number of ascospores per sachet was determined.

8. The experimental design and the repetitions used in the different experiments are not the same. It is difficult to understand the methodology for the different experiments presented, together with 5 diseases, likewise the statistical test used is not the most appropriate, since it takes into account the individuality of the experiments.

This was a large study, and we acknowledge there is a lot to report. We conducted 4 experiments at three locations. Rather than focusing on a single disease, we decided to survey for all disease present to determine efficacy of chitosan against a range of pathogens. We would be interested in more specific feedback from the reviewer about which statistical test is more appropriate. We considered using experiment as a random blocking variable to better determine the overall efficacy of chitosan over multiple locations, however, since treatments varied by location, we decided this was not an appropriate analysis of the data. Thus, analyzing the experiments separately allowed us to examine the efficacy of chitosan treatments on each farm under those unique conditions.

Results

9. It is recommended to attach photos for each of the diseases registered in its evaluation scale, in addition to carrying out a statistical analysis comparing the different varieties of apple trees and delimiting which is the most susceptible.

We have referenced other papers with images of the disease scale used in this study. The goal of this study was to evaluate if chitosan can reduce foliar diseases of apple and was not to determine cultivar susceptibility. As such, this is why we did not compare varieties to report on difference in disease susceptibility.

10. The most vulnerable aspect of the study is that there is no precise identification of the 5 causal agents of the disease, as well as if virulent strains are handled. It is recommended to carry out the molecular identification process to corroborate the taxonomy of the causal agents.

We agree that this would be a great addition to the body of research. However, the overall focus of this research was not on the casual agents but on how chitosan and biopesticides could fit into a disease management system to manage all foliar apple diseases in the northeast. This study was designed to establish 1) if chitosan was effective under these conditions and 2) what future experiments should focus on based on our results. We also relied on natural inoculum, so specific strains were not handled and thus determination of virulence is not applicable to this study. Thus, running causal agent specific testing, including molecular identification, virulence, and in-vitro assays against chitosan are valuable experiments but outside of the scope of this research. However, to address this concern, we added text to describe the diseases detected at each site and how they were identified based on characteristic symptoms with citations.

11. To this is added, that the best treatment presented was chitosan plus a biopesticide or program with a chemical agent. This is counterproductive since the chemical agent penetrates the chitosan membrane and is a potential risk to the health of consumers, in addition to the fact that it can increase the production costs of the apple tree by removing this membrane. However, more studies would be necessary to know the percentage of the product that is present in the apple tree.

We are not aware of published research showing that synthetic fungicides penetrate the chitosan membrane leading to a human health risk. We found one research article reporting on the combined application of chitosan and nanofungicides in which the authors observed a reduction in phytotoxic effects compared to the chitosan or fungicide alone (https://doi.org/10.1371%2Fjournal.pone.0231315).

Future studies should examine the potential of a rotational spray program of chitosan plus a biopesticide and a chemical agent. The reality is that in Northeast apple production, we do not have the scientific advances or the consumer support to completely stop the use of chemical fungicides for managing foliar diseases. Thus, if we can cut down on the number of conventional fungicide applications made throughout the year, we can decrease exposure to the chemicals and decrease costs. This next step should focus on only applying conventional fungicides when there is high risk of infection and utilizing chitosan and biopesticides for coverage the rest of the growing season.

12. Chemicals like Mancozeb have been shown to cause health problems and are linked to cancer problems in farmers.

Yes, that is exactly why we conducted this research and why it is important. Our goal is to aid farmers in decreasing their reliance on conventional fungicides, such as Mancozeb. We included Mancozeb as a control to determine if the biorational treatments have the potential to be just as effective as synthetic fungicides.

13. In the methodology the microscopic observation of ascospores is mentioned and in the results no image is presented, please attach photographic information.

We provided data on ascospore counts as a measure of primary inoculum load in the orchard to compare treatments. We did not feel that images of the spores would aid in interpretation of the effect of chitosan on ascospore production.

Discussion

14. The blue color, go to conclusion.

We revised the discussion and conclusions to reflect this suggestion.

15. Adequate, however, it is recommended to specify the information more, since the authors refer to the disease, but since there are 5 different diseases, in some paragraphs they do differ from the disease, but in most of the discussion they do not, please restructure the justification so that it can be more understandable.

The overall goal of the study was to focus on the effect of chitosan on foliar disease in apple orchards. We specifically do not go into-depth in the discussion about the different diseases as we are not focusing on the interaction between chitosan and each of these pathogens (ie mode of action within the disease cycle). Keeping our discussion focused on the overall efficacy of chitosan keeps the reader focused on the stated objectives.

Conclusion

15. It is recommended to substantially modify the conclusion according to the results presented.

We are not quite sure what changes or suggested improvements are by this reviewer. We modified the section headers in the discussion to state the main take-aways of the research more clearly as it relates to the three objectives. Additionally, edits were made to the conclusion to better articulate the results of the study.

16. Bibliography

Suggested edits to the bibliography have been made

Reviewer 3 Report

The authors have focused on Integration of chitosan and biopesticides to suppress pre-harvest diseases of apple. Interesting paper, a lot of work. Please see my few comments regarding this manuscript:

In the Introduction section authors have used citation clusters, which should be avoided, and authors are desired to use a single citation for one argument or finding. The reference clusters should be split down by presenting the findings of each article independently. Moreover 44 references for a page and a bit of manuscript are much too many. No needing 3-5 references for a statement. Remove the excessive, old, and not relevant references, being added to increase their volume.

L90-94. Aim of the study should be the last, separate paragraph of the Introduction, to be esier visible. Additionally, it is not clear what novelty/special aspects your research brings to the field. Please improve this aspect.

Discussion section. 

 Multiple factors implied in the orchard management must be underlined, to make a complete frame of the topic. I suggest checking and referring to https://doi.org/10.1007/s11356-019-04214-1

Also, importance of nanotechnology should be provided, regarding the aspect of nano-farming versus nanotoxicity – please see https://doi.org/10.1016/j.chemosphere.2021.132533

Before section 5, please add the strengths and the weakness (if any) of your research, in a LAST paragraph of Discussion.

Minor editing errors should be corrected.

Author Response

1. In the Introduction section authors have used citation clusters, which should be avoided, and authors are desired to use a single citation for one argument or finding. The reference clusters should be split down by presenting the findings of each article independently. Moreover 44 references for a page and a bit of manuscript are much too many. No needing 3-5 references for a statement. Remove the excessive, old, and not relevant references, being added to increase their volume.

While we agree that citation clusters are not always beneficial, it is important to give proper acknowledgement of the research that has been done through referencing multiple review articles. We have made edits so that when listing specifics, the citations are after the proper component. For example: “Several natural compounds, such as cellulose [13], chitin [14], and chitosan [15,16], have shown a synergistic effect on biopesticide efficacy for reducing plant diseases.”

2. L90-94. Aim of the study should be the last, separate paragraph of the Introduction, to be esier visible. Additionally, it is not clear what novelty/special aspects your research brings to the field. Please improve this aspect.

We agree and have made the objectives a separate paragraph in the introduction. Additionally, two sentences were included to highlight the importance of this research.

Discussion section. 

3. Multiple factors implied in the orchard management must be underlined, to make a complete frame of the topic. I suggest checking and referring tohttps://doi.org/10.1007/s11356-019-04214-1

Thank you for this suggestion. Climate change can lead to the establishment of new pathogens or changes in pathogen infection cycles. While the effect of environment on apple production was not the topic of this study, we agree that researching new crop protection tools, such as chitosan, will help growers adapt to climate driven changes in pathogen outbreaks. A line has been added to the discussion to address this point.

4. Also, importance of nanotechnology should be provided, regarding the aspect of nano-farming versus nanotoxicity – please see https://doi.org/10.1016/j.chemosphere.2021.132533

We agree that nanotechnology is an important aspect of current agricultural research; however, that is largely outside of the scope of our research. A sentence was added to state that there are other products and technologies, such as chitosan nanoparticles, that could be studied for foliar disease management.

5. Before section 5, please add the strengths and the weakness (if any) of your research, in a LAST paragraph of Discussion.

We discussed weaknesses in the research throughout the discussion when it was appropriate to highlight where future research should focus. We feel that it would be repetitive to repeat these comments in a separate paragraph.

Reviewer 4 Report

Congratulations for your article ! 

Author Response

How original is the topic? What does it add to the subject area compared with other published material?

The approach does not present originality, studies on the pre-harvest use of chitosan can be found in the specialized literature (doi:10.17265/2162-5298/2017.11.004).

We agree that there are other research articles that report on pre-harvest application of chitosan to manage a range of diseases, however only one other study has examined pre-harvest application of chitosan in an apple orchard and thus we feel this research is novel and adds to the literature. Additionally, many studies have reported on reagent grade chitosan (not available to farmers) and very few studies have examined the efficacy of commercially available chitosan products for managing disease.

Round 2

Reviewer 1 Report

horticulturae-2425957-v2: It is very nice work and thank you very much for the revised manuscript. However, there are still some points have not revised.

1) Lines 29-31: The Refs. [3] and [4] did not investigate how synthetic pesticides impact to human health and environment. Thus, these papers must be included to support in terms of the impacts on human health and environment. [Environmental, Human and Ecotoxicological Impacts of Different Rice Cultivation Systems in Northern Thailand. Int. J. Environ. Res. Public Health 2023, 20(3), 2738] [Carbon, Nitrogen and Water Footprints of Organic Rice and Conventional Rice Production over 4 Years of Cultivation: A Case Study in the Lower North of Thailand. Agronomy 2022, 12(2), 380.]. 

2) Mancozeb is one of hazard fungicide. Why it was chosen in the treatment? Why not selected the other type? Please provide more explanation.

3) Figure 2: Letters a-c must be provided the meaning. Although you mentioned “same letter”, it is unclear which letters. Same comment for Figure 3, Tables 5, 6, 7, and 8.

4) Section “4. Discussion” needs the sub-topics as follow:

      4.1 Chitosan can reduce disease when applied as part of a conventional fungicide program

     4.2 Synergisms between chitosan and biopesticides varied by site, cultivar and pathogen

     4.3 Chitosan did not reduce overwintering spores of V. inaequalis

-

Author Response

1) Lines 29-31: The Refs. [3] and [4] did not investigate how synthetic pesticides impact to human health and environment. Thus, these papers must be included to support in terms of the impacts on human health and environment. [Environmental, Human and Ecotoxicological Impacts of Different Rice Cultivation Systems in Northern Thailand. Int. J. Environ. Res. Public Health 2023, 20(3), 2738] [Carbon, Nitrogen and Water Footprints of Organic Rice and Conventional Rice Production over 4 Years of Cultivation: A Case Study in the Lower North of Thailand. Agronomy 2022, 12(2), 380.]. 

We agree that the abovementioned references are useful, however, they are focused predominately on rice production. Our references are reviews that address broader impacts of pesticides throughout the agricultural system. Reference [3] reviewed what is known about the impacts of synthetic pesticides on human health in the third section of their paper under the heading “human health effects of pesticide use”. While reference [4] discussed the environmental impacts of pesticides, we have also included an additional reference that supports this statement (DOI: 10.1080/07352689.2011.554355).

2) Mancozeb is one of hazard fungicide. Why it was chosen in the treatment? Why not selected the other type? Please provide more explanation.

We selected Mancozeb as a control treatment as it is a commonly used conventional fungicide for management of foliar apple diseases, such as apple scab, in the Northeastern United States. This fungicide represented the “grower standard” to which any alternative product will need to be compared in terms of efficacy in reducing foliar diseases. The hazardous nature of Mancozeb is why we are doing this research, which is to reduce the use of this chemical in apple orchards…but we must demonstrate to growers that softer products can be just as effective.

3) Figure 2: Letters a-c must be provided the meaning. Although you mentioned “same letter”, it is unclear which letters. Same comment for Figure 3, Tables 5, 6, 7, and 8.

This notation is common practice in Table and Figure captions in the scientific literature and is used to denote statistically significant differences (when two letters are different) OR to indicate where the is no-significant difference (letters are the same). Examples of other articles using this methodology can be found at the following links: https://doi.org/10.3390/horticulturae9010096; https://doi.org/10.1016/j.scienta.2015.02.014; https://doi.org/10.3390/horticulturae9060673

4) Section “4. Discussion” needs the sub-topics as follow:

      4.1 Chitosan can reduce disease when applied as part of a conventional fungicide program

     4.2 Synergisms between chitosan and biopesticides varied by site, cultivar and pathogen

     4.3 Chitosan did not reduce overwintering spores of V. inaequalis

We have made these edits within the text.

Reviewer 2 Report

The authors performed the vast majority of the recommended observations and clarified doubts about the work. I consider that it has improved substantially and that it can be published.

Author Response

The authors performed the vast majority of the recommended observations and clarified doubts about the work. I consider that it has improved substantially and that it can be published.

Thank you for the review and your recommendation to accept for publication. No additional edits were requested and thus there are no specific responses to this review.